# Mesoporous Silica Modified with 2-Phenylimidazo[1,2-a] pyridine-3-carbaldehyde as an Effective Adsorbent for Cu(II) from Aqueous Solutions: A Combined Experimental and Theoretical Study

**DOI:** 10.3390/molecules27165168

**Published:** 2022-08-13

**Authors:** Rafik Saddik, Imad Hammoudan, Said Tighadouini, Othmane Roby, Smaail Radi, Maha I. Al-Zaben, Abir Ben Bacha, Vijay H. Masand, Zainab M. Almarhoon

**Affiliations:** 1Laboratory of Organic Synthesis, Extraction and Valorization, Faculty of Sciences Ain Chock, Hassan II University, Casablanca 20000, Morocco; 2Laboratory of applied Chemistry and Environment (LCAE), Faculty of Sciences, Mohamed Premier University, Oujda 60000, Morocco; 3Department of Chemistry, College of Science, King Saud University, P.O. Box 2455, Riyadh 11451, Saudi Arabia; 4Biochemistry Department, College of Science, King Saud University, P.O. Box 22452, Riyadh 11495, Saudi Arabia; 5Laboratory of Plant Biotechnology Applied to Crop Improvement, Faculty of Science of Sfax, University of Sfax, Sfax 3038, Tunisia; 6Department of Chemistry, Vidya Bharati Mahavidyalaya, Amravati 444 602, India

**Keywords:** heavy metals, adsorption, silica functionalization

## Abstract

In this study, we will present an efficient and selective adsorbent for the removal of Cu(II) ions from aqueous solutions. The silica-based adsorbent is functionalized by 2-phenylimidazo[1,2-a] pyridine-3-carbaldehyde (SiN-imd-py) and the characterization was carried out by applying various techniques including FT-IR, SEM, TGA and elemental analysis. The SiN-imd-py adsorbent shows a good selectivity and high adsorption capacity towards Cu(II) and reached 100 mg/g at pH = 6 and T = 25 °C. This adsorption capacity is important compared to other similar adsorbents which are currently published. The adsorption mechanism, thermodynamics, reusability and the effect of different experimental conditions, such as contact time, pH and temperature, on the adsorption process, were also investigated. In addition, a theoretical study was carried out to understand the adsorption mechanism and the active sites of the adsorbent, as well as the stability of the complex formed and the nature of the bonds.

## 1. Introduction

Environmental pollution has become a major concern for mankind and the eco-system. It has impacted human health and it has led to economic and social issues for millions of people worldwide. Furthermore, water pollution is considered one of the most serious problems that requires an immediate and radical solution. It could also lead to serious threats to sustainable environmental development [1]. Rapid industrial development in many countries and human activities result in the contamination of water with heavy metal ions like Cu, Pb, Hg, etc., and this problem has caused several diseases [2]. Cu(II) is toxic at high concentrations and responsible for numerous illnesses due to its bio-accumulable and non-biodegradable nature [3]. Several techniques such as adsorption [4], liquid–liquid extraction [5], membrane filtration [6], ion exchange [7] and electrolysis [8] have been used for the purification of wastewater containing heavy metal ions. Recently, adsorption has emerged as a more promising technique due to: (i) ease of operation, (ii) higher enrichment factor, (iii) reduction of organic solvent usage, low cost and extraction time, (iv) high selectivity, (v) ease of separation and (vi) ability to combine with different modern detection techniques [9]. Various types of materials such as silica [10], activated carbon [11], biopolymers [12], zeolite [13] and clay [14] are successfully employed as effective adsorbents. Compared with other substrates, silica gel **has garnered attention** in sorption of heavy metals due to its large surface area and high thermal and mechanical stability [15]. Much of the research in recent times has reported the immobilization of organic ligands onto mesoporous silica to decontaminate heavy metal from wastewater [16]. The efficiency of these adsorbents depends mainly on the affinity of the donor atoms (S, O and N) that are deposited on the surface of the materials. Several ligands, bearing donor atoms, have been synthesized and immobilized on the surface of silica for water pollution control [17,18,19,20]. In this context, imidazo[1,2-a] pyridine has attracted attention because it plays a major role in coordination chemistry [21,22]. On the other hand, the condensation of an amine of silica modified with an aldehyde group of imidazo[1,2-a] pyridine leads to the formation of a Schiff base, which is known to have high bonding affinities and to form very stable complexes with metal ions [23,24].

In the present work, imidazo[1,2-a] pyridine-3-carbaldehyde has been used to functionalize silica for the first time and to afford a functional adsorbent. In fact, its structure was fully characterized. The new synthetic material shows a high selectivity towards Cu^2+^ ions, it also shows a high adsorption capacity compared to other similar adsorbents [9]. The influence of pH value, contact time, metal concentration and temperature for the adsorption of Cu(II) were studied using the batch method. In addition, the selectivity and regeneration capacity were discussed, and the adsorption mechanism was revealed on the basis of theoretical calculations.

## 2. Experimental Section

### 2.1. Materials and Methods

All reagents (Aldrich, purity 99.5%) were of analytical grade. Initially, silica gel (60 Å, 70−230 mesh) (E. Merck) was activated at 120 °C for 24 h. The quantification of metal ions in aqueous solutions was determined by atomic absorption (Spectra Varian A.A. 400 spectrophotometer, Shelton, CT, USA). The surfaces were characterized by a CHN analyzer (Microanalysis Center Service, CNRS, Paris, France), Fourier-transform infrared spectroscopy (FTIR, Perkin Elmer System 2000, Waltham, MA, USA) at 25 °C, SEM (FEI-Quanta 200, Hillsboro, OR, USA), TG/DTA (Perkin Elmer Diamond, Waltham, MA, USA) under a 90:10 oxygen/nitrogen atmosphere at 10 °C·min^−1^, 13C solid state nuclear magnetic resonance (NMR, CP MAX CXP 300 MHz, Lille, France) and BET (ThermoQuest Sorpsomatic 1990 analyzer, Lille, France). Nitrogen adsorption–desorption isotherm plots were obtained on a Thermoquest Sorpsomatic 1990 analyzer after the materials had been purged in a stream of dry nitrogen. The pH determinations were carried out with a pH 2006, J. P. Selecta s. a. pH meter.

### 2.2. Synthesis of 2-phenylimidazo[1,2-a] pyridine-3-carbaldehyde

2-Phenylimidazo[1,2-a] pyridine (3 mmol) was suspended in CHCl_3_, the obtained solution was added to a mixture of POCl_3_ (11 mL) and DMF (8 mL) at 0–5 °C, under stirring for 30 min, followed by refluxing for 6 h and then neutralized using Na_2_CO_3_. The residue was extracted by dichloromethane and purified on silica gel to give a white solid.

^13^C NMR (100 MHz, CDCl_3_) δ (ppm): 115.10 (C6), 117.22 (C8), 120.62 (C3), 128.36 (C7), 128.92 (2Cph), 129.45 (3Cph), 130.18 (C5), 132.32 (Cph), 147.51 (C8a/C2), 157.90 (C2/C8a), 179.26 (CHO)

### 2.3. Preparation of Amine-Functionalized Silica SiNH2

A mixture of activated silica (10 g) and dry toluene (50 mL) was stirred under reflux for 2 h. After that, 3-aminopropyltrimethoxysilane (10 mL) was added dropwise to the suspended solution and the refluxing was maintained for 24 h. The **SiNH2** formed was filtered and washed by Soxhlet extraction with different organic solvents for 12 h. Finally, the obtained modified silica was dried at 60 °C for 24 h.

### 2.4. Fabrication of the **SiN-imd-py** Adsorbent

To prepare the (**SiN-imd-py**), a mixture of 2-phenylimidazo[1,2-a]pyridine-3-carbaldehyde (0.5 g) and 3- aminopropylsilica (**SiNH2**) (3.5 g) in 30 mL of methanol was mixed, heated at reflux (140 °C) and stirred for 24 h. At room temperature, the solid residue was filtered. Methanol, dichloromethane, THF and diethyl ether were used in Soxhlet extraction of the product for 10 h. The desired solid product was dried completely (Figure 1).

### 2.5. Batch Adsorption Experiment

The parameters influencing the adsorption of metal ions via the material, such as the effect of concentration (10 to 300 mg L*^−^*^1^), contact time (5 to 35 min), pH (1 to 7) and temperature (25 to 45 °C), were studied by the batch method. For this purpose, 10 mL of the ionic solution containing the following salts: (Pb(NO_3_)_2_,6H_2_O); (Cd(NO_3_)_2_, 6H_2_O); (Cu(NO_3_)2,3H_2_O) and (Zn(NO_3_)2,6H_2_O), together with 10 mg of hybrid material, is introduced into a set of test tubes. The pH values are adjusted with diluted HCl and NaOH solutions, and the mixture is stirred at 25 °C. The adsorption capacities are determined as follows [25]:q_e_ = (C_0_ − C_e_) × V/m(1)
where q_e_ (mg g*^−^*^1^) is the adsorption capacity; C_0_ and C_e_ (mg L^−1^) are, respectively, the initial and equilibrium concentrations; m(g) is the mass of the adsorbent; and V (L) is the volume of the solution.

### 2.6. Computational Methods

Density functional theory (DFT) [26] calculations and QTAIM were used to illustrate the adsorption mechanism [27]. The Gaussian 09 [28] software and gaussview05 [29] were used to optimize the structure. These software and methods were used to investigate the interaction between the material and metal ion Cu^2+^ [30]. All the structures reported herein were fully optimized in gas phase at the B3LYP/GenECP level of theory [31,32]. The LanL2DZ relativistic pseudo potential [33] basis set was used for the metals, and the 6–311++G (d, p) [34,35] basis set was used for all other atoms. The interaction energy between a single **SiN-imd-py** hydrogel unit and metal ions (Cu^2+^) was calculated by:ΔIE = E_(complex)_
*−* (E_material_ + E_ion_)(2)

These findings were supported by one of the most well-known DFT indices [36], the nucleophilic Parr functions (P-) [37], which is a useful tool for assessing the potential nucleophilic compound sites. By adding an electron of the Mulliken atomic spin density at the radical cation, we obtain these functions. Bader proposed the technique of topology analysis for analyzing electron density in “atoms in molecules” (AIM) theory, which is also known as “quantum theory of atoms in molecules” (QTAIM) [38]. The electron localization function ELF [39] is used for finding the electron density. Because ELF is a density-based function that can be understood in terms of the relative local excess of kinetic energy density, it has been recognized as a technique that provides a good framework for the investigation of changes in pair electron density [40]. The localized-orbital locator (LOL) [41] is used to further understand the physical characteristics of the chemical bond between two atoms. Schmider and Becke [42] have recently presented a crucial tool, the localized-orbital locator, to explain bonding properties in terms of the local kinetic energy. The latter is roughly defined as a function of electron density and its first and second derivatives.

## 3. Results and Discussion

### 3.1. Characterization

#### 3.1.1. Elemental Analysis

Elemental analysis was used to assess the elemental composition of the materials and to confirm the presence or absence of an organic motif on the modified silica **SiNH2**. (Table 1)

Comparing w.t% of the two materials **SiNH2** and **Si-imd-py**, we notice an increase in the percentages of the elements of C, N and H, which indicates that the functionalization was successfully realized.

#### 3.1.2. FTIR

FTIR characterization technique allowed us to identify the different types of vibration of the functional groups presented in the different materials studied. FTIR spectra of **SiG**, **SiNH2** and **SiN-imd-py** materials are shown in Figure 1. Indeed, the spectrum of the free silica SiG shows a large band at 3446 cm*^−^*^1^ and a weak band at 1652 cm*^−^*^1^ attributed to O-H elongation vibrations. The bands at 1086 and 798 cm*^−^*^1^ are attributed to Si-O-Si and Si-O vibrations, respectively [43]. For the **SiNH2** spectrum, two new adsorption bands were observed around 2932 and 1582 cm*^−^*^1^, which are characteristic of C-H and NH2 elongation vibration resettlement [44]. These results indicate that the spacer arm has been immobilized on the silica surface. For the spectrum of the **SiN-imd-py** material, we also note the appearance of a new band at 1440 cm*^−^*^1^ that is attributable to the C=N vibration, which confirms the grafting of the organic motif onto the **SiNH2** support.

#### 3.1.3. Scanning Electron Microscope (SEM)

Scanning electron microscopy (SEM) allows the observation of the morphology of material and the distribution of grains on the material surface. The micrographs obtained are shown in Figure 2. As we can see from the SEM images, the morphology of the non-functionalized and functionalized silica particles is different. An irregular and smooth surface was observed for **SiG,** while the morphology of **SiNH2** and **SiN-imd-py** materials is rough and porous, and agglomerations of molecules appear on the surface of the materials, indicating the presence of organic molecules on the silica surface.

#### 3.1.4. Thermogravimetric Analysis (TGA)

Thermogravimetry makes it possible to follow, as a function of temperature, the evolution of the loss of mass of the sample, mainly caused by the vaporization of water and by the destruction of organic matter. Figure 3 shows the thermogravimetric curves of **SiN- imd-py** material and both materials, **SiNH2** and SiG. For virgin silica, two stages of mass loss can be distinguished. The first stage that corresponds to a weight loss is at 3.15% (between 25 °C and 110 °C). It was assigned to the desorption of water adsorbed on the surface of silica gel. The second stage of mass loss is at 5.85% observed between 110 °C and 800 °C and is attributed to the condensation of silanol groups leading to siloxane groups (**Si-O-Si**). For the **SiNH2,** it presented a high mass loss at 6.65% between 200–800 °C. This mass loss is related to the decomposition of the organic chain immobilized on silica. The **SiN-imd-py** material is also presented in two decomposition stages. The first loss stage, between 25 and 110 °C, is at 4.48%. This is attributable to the removal of adsorbed water. The second mass loss between 200 to 800 °C is 16.68%, due to the degradation of the organic fragment grafted on the silica gel surface.

#### 3.1.5. N2 Physisorption Studies

To investigate the changes in surface area and porosity of **SiN-imd-py** material, we measured the specific surface area (SBET) according to the Brunauere–Emmett–Teller method [45] and the pore volumes were measured according to the Barret–Joyner–Halenda (BJH) method [46]. Figure 4 shows that the material presents a type IV isotherm, which clearly indicates that the material is mesoporous according to the classification of the International Union of Pure and Applied Chemistry. The figure also reveals the H2-type hysteresis loop for partial pressures P/P0 ≥ 0.4. The results of the physical properties found are reported in Table 1.

From the analysis of the data in Table 2, we observe that the **SiN-imd-py** material is characterized by a decrease in pore volume and specific surface area compared to the **SiNH2** precursor. This decrease is due to the grafting of organic ligands on the silica surface which partially blocks the adsorption of nitrogen molecules.

### 3.2. Adsorption Studies

#### 3.2.1. Effect of pH

The pH of the solution is one of the most important parameters controlling the adsorption of metal ions. In this study, the effect of various pH values (1–7) at 25 °C was investigated by means of fabricated functionalized material. As shown in Figure 5, when the pH is below 5, the adsorption capacity increases with the pH; when the pH > 5, the adsorption capacity of the adsorbent tends to be stable. The SiN-Imd-Py adsorbent has a higher adsorption capacity for Cu(II) over a wider pH range. Therefore, modification of the imidazopyridine on the SiNH2 surface can significantly improve the adsorption capacity. When the pH value was at 1–2, the adsorbent **SiN-imd-py** showed a negligible amount of adsorbed Cu(II). At this moment, the grafted imidazopyridine–Schiff base ligand gradually deprotonated, and this implies that the positive charge increases, the electrostatic repulsion with the metal cation increases, and the adsorption performance decreases. The adsorption of Cu(II) on the adsorbent is difficult to quantify at pH above 7 because it can be masked by the precipitation phenomenon in the Cu(OH)_2_ form. The adsorption capacity of Cu(II) ion reached its maximum at a pH between 6 and 7. Therefore, a pH of 7 was chosen for the extraction of the Cu(II) studied in all subsequent studies.

#### 3.2.2. Effect of Contact Time and Adsorption Mechanism

The effect of time is of considerable practical interest in adsorption and one of the most important characteristics defining the effectiveness of an adsorption. The adsorption kinetics for Cu(II) onto the **SiN-imd-py** are shown in Figure 6. It can be clearly seen that the adsorption of Cu(II) is very rapid during the first five minutes. This could be attributed to the large number of ligands available for chelation and a high level of free Cu(II) ions at this stage. After 30 min, the sorption rate decreased and finally reached a plateau, thus confirming the rapid adsorption kinetics for Cu(II). Equilibrium would have been reached due to saturation of the adsorption sites. Consequently, the optimal contact time was confirmed at 30 min for the following experiments.

To better comprehend the adsorption procedure, it is necessary to study the adsorption kinetics. This procedure is evaluated by two models, pseudo-first-order and pseudo-second-order, which are generally interpreted by the following equations [47,48]:

Pseudo-first-order model:ln(q_e_ − q_t_) = lnq_e_ − k_1t_
(3)

Pseudo-second-order model:t/q_t_ = 1/K_2_q_e_^2^ + t/q_e_(4)
where q_e_ (mg g*^−^*^1^) and q_t_ (mg g*^−^*^1^) are the amounts of adsorbate at equilibrium and at time t (s), respectively. K_1_(min*^−^*^1^) and K_2_ (g (mg min*^−^*^1^)*^−^*^1^) are the rate constants of the first-order and the second-order models, respectively. The corresponding parameters for the two kinetic models are listed in Table 2. The theoretical qe value of the pseudo-second-order model fits the experimental values better, indicating that the model is proposed to describe the adsorption kinetics of Cu(II) by the **SiN-imd-py** adsorbent (Figure 7). The correlation coefficient (R²) of the pseudo-second-order kinetic model is near to 1, much higher than the correlation coefficient of the pseudo-first-order model. The sufficiency of the experimental data with this model indicates that Cu(II) adsorption is controlled by the chemical procedure (chemisorption) due to the existence of chemical interactions between the metal and the ligand grafted on the silica (coordination bonds).

#### 3.2.3. Influence of Initial Concentration

In the adsorption process, the effect of concentration plays an important role because it provides necessary information about adsorbing performance in different concentrations. Accordingly, different initial concentrations of Cu(II) varying from 10 to 300 mg L^−1^ were employed to investigate the adsorbents further using the batch method. As shown in Figure 8, the amount of Cu(II) adsorbed by the **SiN-imd-py** gradually increases with increasing initial metal ion concentration. This suggests that the available adsorption sites on the adsorbent surface were handily occupied by the metal ions and that the adsorption capacity was very high. As the initial metal concentration increases, a saturation value is reached. This observation can be explained by the fact that there are fewer active sites on the surface of the adsorbent that are available for further adsorption of metal ions. This result suggests that the chelation of the Imidazopyridine-Schiff base heightens the adsorption capacity of the adsorbent.

#### 3.2.4. Adsorption Isotherms

The equilibrium relationships between the concentration of the adsorbate and the amount of adsorbate accumulated on the adsorbent were explained by the adsorption isotherms. Indeed, the adsorption capacity results were fitted by four classical isotherm models, namely Langmuir, Freundlich, Dubinin-Radushkevich (D-R) and Temkin, which are generally described as the following equations [49,50]:

Langmuir model:C_e_/q_e_ = C_e_/q_m_ + 1/qK_L_
(5)
where C_e_ (mg L*^−^*^1^) is the equilibrium concentration, q_e_ and q_m_ (mg g^−1^) denote the equilibrium and theoretical maximum adsorption of the adsorbent, K_L_ (L. mol*^−^*^1^), is the Langmuir affinity.

Freundlich model:q_e_ = K_F_ Ce^1/n^(6)
where K_F_ is the Freundlich constant and n is energy or intensity of adsorption.

Dubinin–Radushkevich (D-R) model:ln (q_e_) = ln (q_m_) − βε^2^
(7)
where β (mol J*^−^*^1^)2 is a constant related to the mean free energy E of adsorption (E = (2β) − 0.5), and ε (J mol*^−^*^1^) is the Polanyi potential related to the equilibrium concentration (ε = RT ln (1 + 1/C_e_)).

Temkin model:q_e_ = RT/b_t_ lnA_t_ + RT/b_t_ ln C_e_
(8)
where b_t_ (J mol*^−^*^1^) and At (L mg*^−^*^1^) are Temkin isotherm constant and T defines the Kelvin temperature (K), and R (8.314 J mol*^−^*^1^ K*^−^*^1^) defines the universal gas constant.

The fitting plots of the Langmuir model are presented in Figure 9, and the theoretical parameters of four adsorption isotherms, as well as the regression coefficients, are listed in Table 3. From the values in Table 4, the Langmuir model provides good correlation coefficient (>0.990), suggesting that the Langmuir model was appropriate to describe the adsorption process of Cu(II) onto **SiN-imd-py** adsorbent. On the other hand, the calculated equilibrium adsorption capacity q (mg g*^−^*^1^) based on the Langmuir model is much closer to experimental data qe (mg g*^−^*^1^), which also proved the adequacy of the Langmuir model. The above fact demonstrated that the adsorption process of Cu(II) by the adsorbent was attributed to a homogeneous monolayer adsorption and the maximum adsorption capacity was examined at 103.51 mg g*^−^*^1^.

#### 3.2.5. Thermodynamic Studies

The thermodynamic behavior was studied to determine the thermodynamic parameters of Cu(II) adsorption onto **SiN-imd-py**, including the change in standard enthalpy (ΔH°), standard entropy (ΔS°) and Gibbs energy (ΔG°), which were determined using the following equations [51]:K_d_ = (C_0_ − C_e_)/C_e_
(9)
Ln K_d_ = ΔS°/R − ΔH°/RT (10)
ΔG° = ΔH° − TΔS° (11)
where K_d_ implies the adsorption distribution coefficient. R (8.314 J mol*^−^*^1^ K) and T(K) are the universal gas constant and the absolute temperature, respectively. The curve Ln(K_d_) versus 1/T (Figure 10) was employed to determine the thermodynamic parameters ΔG°, ΔH° and ΔS° which are given in Table 5. The positive value of ΔH° provides clear evidence that the adsorption phenomenon was endothermic. The endothermic nature also involves the absorption of energy in the form of heat from the environment during the adsorption process. The positive values of ΔS° indicate an increase in randomness at the solid–liquid interface during the adsorption of Cu(II) ions, mainly due to the dehydration of metal ions towards the adsorption sites. The negative ΔG° values at all temperatures indicate that the adsorption phenomenon is spontaneous and more favorable at high temperatures. The metal ion adsorption process suggests that a large amount of heat is consumed to transfer the metal ions from the solution to the solid phase. The thermodynamic results show that the elimination of Cu(II) on the adsorbent was endothermic and spontaneous in nature.

It is clear from this table that all the values of ΔG° are negative. The results presented above suggest that the adsorption process involved is spontaneous. On the other hand, the change of the standard free energy decreases with increasing temperatures regardless of the nature of adsorbent. This indicates that a better adsorption is actually obtained at higher temperatures.

#### 3.2.6. Adsorption Selectivity for Cu(II)

The competitive adsorption of heavy metal ions by **SiN-imd-py** material was studied in a mixed quaternary system (Cu(II), Zn(II), Cd(II) and Pb(II)) by the batch method (Figure 11). The material has a higher selectivity towards Cu(II) compared to other ions. The high selectivity of **SiN-imd-py** towards Cu(II) is due to the types of ligand immobilized on the silica surface, reflecting the extraordinary ability and efficiency to form more stable complexes with Cu(II).

#### 3.2.7. Desorption and Recycling

Desorption is one of the most important indicators to evaluate whether an adsorbent is practical or not. Adsorbent desorption was examined by adding 2 mol/L HCl to 10 mg of material-Cu(II) with stirring at room temperature for 60 min. Then, the material was filtered and neutralized by a NaOH solution. After washing, the material was dried with the help of a vacuum desiccator and then in the oven for the next adsorption. Table 6 represents the extraction efficiency of the material after five cycles of Cu(II) adsorption–desorption. It is interesting to note that the adsorbents retained more than 95% of their adsorption capacity. Thus, the material has an exceptional recyclability and applicability that could be useful for the purification of copper-contaminated wastewater.

#### 3.2.8. Comparison with Similar Adsorbents

Table 7 shows a comparative study highlighting the interest and efficiency of our adsorbent towards Cu(II) compared to others recently described in the literature.

### 3.3. Adsorption Mechanism

#### 3.3.1. Active Sites Study

The nucleophilic indexes of Parr (P-) allow us to determine the atomic sites of the donor species. For the material **SiN-imd-py**, one nitrogen N2 (Figure 12) atom is not considered to be a reactive site because the value of (P-) is negative −0.064 (Table 8). This indicates that it is not involved in coordination with Cu. Nevertheless, the nitrogen N9 and N11 display positives value of p-, revealing that these are suitable sites for an electrophilic attack via Cu^2+^ ions.

The QTAIM approach is necessary to differentiate between the two nucleophilic sites N9 and N11; as shown in Figure 12 (right), the N9 site (the red arrow) is favored to attack the copper ion Cu^2+^. By NCI we could confirm the attack of N9 because from the left Figure 12 (the red arrow), no interaction was observed in this region, which further favors the attack of N9 on this side of the molecule [36].

#### 3.3.2. Cu(II) Complexation Study

The optimized geometries of possible complexes are proposed on the basis of NBO analysis, p- indices and QTAIM. In this sense, E was calculated, which is the interaction energy; this was −229,208 for complex1 and −229,199 for complex2 which means that complex1 is more stable in relation to complex2 [61].

To obtain information on the binding nature of the studied complexes we introduced the analysis of the natural binding orbital NBO [62]. The interaction between the N9, O39, O42 and O45 atoms was evaluated by the second order energy E(2) and the electronic configuration (EC) (Table 9). The results of the NBO analysis presented in the table reveal that the four atoms N9, O39, O42 and O45 have almost the same EC, but for Cu^2+^ this is not the case (EC(Cu^2+^(1)) < EC(Cu^2+^(2)).

The secondary energy E(2) values are also listed in the same table, and we see that all energy values found for complex1 are higher than those found for complex2. This shows that the charge transfer is from N9, O39, O42, and O45 to the empty orbitals of Cu^2+^, and also indicates that complex1cis has undergone an important electron transfer.

Regarding the bond lengths, they are important to determine the type of bond between the atoms as well as the possibility of existence of a bond, because if the bond length is large enough, an adequate interaction between the two atoms cannot take place. The results displayed in Table 10 show that the distance between Cu and other atoms is small (approximately 1.983 Å to 2.249 Å). This fact shows that the interaction between Cu and other atoms is possible. Lower values of the same distance for complex1 show that it is favorable as a stable complex. QTAIM was the most innovative work of the Bader Group, and an interatomic contour line with a (3,–1) cp (Figure 13) implies that electron density is accumulated among the nuclei that are reassembled in this way. The corresponding values of electron density and lagrangian for complex1 (0.832, 0.510) are large compared to complex2 (0.808, 0.129), while the results of the calculation of the laplacian for complex1cis 0.135 are small compared to complex2 0.495; this means that the bonds formed for complex1 are strong compared to complex2. Consequently, the stability of complex1 is important [62].

The ELF localized electron function is recognized as a technique offering a suitable framework for the study of electron density changes. Similarly, Schmider and Becke proposed the localized-orbital locator to define the characteristics of bonding in terms of local kinetic energy. The results (Table 10) show that the ELF and LOL values for complex1cis (respectively 0.102 and 0.252) are higher than the values for complex2 (0.100, 0.250) which reflects the higher stability of complex1.

To highlight the properties of a line, we have chosen to draw this curve (Figure 14), where the dotted line corresponds to the position Y=0 and the red curve represents the position of the two selected atoms Cu and N. For the complex1trans (left), the two values (1.345 and 5.417) indicate the two atoms that we have chosen; also for complex1 (right) the atoms are located at 1.350 and 5.399 according to the Bohr radius.

It is noticeable that there is a small difference between the two curves. For complex1, the peak at position 1.345 rose to 0.009, but the peak at 1.350 of the other complex1 changed to 0.016. This explicitly explains the difference in stability between the two complexes. In Figure 15, the contour maps of electron density on the plane (N9-Cu-O39) for the two complexes show that there are simple bonds between these three atoms. The same remark can be made for the plan maps of the localized-orbital locator in Figure 15 on the right-hand side.

## 4. Conclusions

In conclusion, we have developed a new hybrid material based on silica and heterocyclic ligand. This material was prepared by immobilization of 2-phenylimidazo[1,2-a] pyridine-3-carbaldehyde on the silica surface. In addition, the synthesized material was characterized by the usual physicochemical techniques (AE, IR-TF, SEM, BET and ATG). The **SiN-imd-py** material showed a marked adsorption efficiency towards Cu(II) from wastewater (97.11 mg g^−1^). We also identified the different parameters influencing the adsorption such as pH, contact time, initial concentration and temperature. Imidazopyridine-Schiff base ligand is very sensitive to pH changes **as** the optimal pH for adsorption is at pH = 6. The equilibrium adsorption time was about 30 min. The kinetic study shows that the adsorption follows the pseudo-second-order model, indicating that the adsorption is chemical. The study of adsorption isotherms is described by the Langmuir model, which shows that the adsorption is monolayer. Actually, the thermodynamic parameters indicate that the metal adsorption process is spontaneous and endothermic. The NCI method and the Parr p-indices help to determine the nucleophilic sites of the material, as in our work. Thanks to these methods, we could find the active sites of **SiN-imd-py**. To highlight the stability of the complexes and the nature of the bonds, we used the following methods: DFT, NBO, QTAIM, ELF and LOL. Clearly, by means of these methods, we found that complex1 is more stable. It is interesting to note that the adsorbent shows selectivity towards Cu(II). This material is easily regenerated by acid washing. Finally, the adsorbent is very promising for the purification and extraction of Cu(II) from wastewater.

## Data Availability

Not applicable.

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
