# Peer review of "Mesoporous Silica Modified with 2-Phenylimidazo[1,2-a] pyridine-3-carbaldehyde as an Effective Adsorbent for Cu(II) from Aqueous Solutions: A Combined Experimental and Theoretical Study"

_molecules, 2022, doi:10.3390/molecules27165168_

Round 1
Reviewer 1 Report
The main target of the current work is to investigate the mesoporous silica modified with 2 phenylimidazo[1,2-a] pyri- 2 dine-3-carbaldehyde ( SiN-imd-py)as an adsorbent for Cu(II) from an aqueous solution. Authors discussed the functionalization of SiN-imd-py using several chemical, physicochemical and computational techniques. The investigated material presents remarkable adsorption efficiency towards Cu(II) from wastewater. The present reviewer found that it is a worthwhile manuscript that deserves publication, but the manuscript suffered from grammatical errors as well as ambiguous words and sentences. Thus, I recommend "Minor revision" for this manuscript.
Some specific comments are presented below.
(1) Abstract
Abstract should be concise and the authors need to improve with more specific short results.
(2) Keywords
Keywords should be revised and improved.
(3) Introduction
The introduction section should be modified though citing recent references related studies and indicating the novelty of the study compared to the carried works.
(4) Analysis of experimental results
· More details regarding the main experimental results, such as surface characterization and adsorption mechanism sections, must provide in the text.
· All Abbreviations should be defined.
· All Figures should be improved due to the low quality in the manuscript.
· Quantum chemical results should be revised and each considered parameter needs its own unity.
(5) General expression issues
The authors ought to go through the manuscript again to eliminate ambiguous expressions which were resulted from complicated sentence structure and grammatical errors.
Author Response
Dear professor,
The authors would like to thank for their efforts in handling this scientific word. The interesting remarks are gratefully acknowledged and will improve the quality of this work.
All comments and suggestions have been addressed in the new version of the manuscript.
Point (1) : Abstract should be concise and the authors need to improve with more specific short results.
Response 1: this is done. It was reduced
Point 2 : Keywords should be revised and improved.
Response 2: This is done. Keywords has been revised.
Point 3: The introduction section should be modified though citing recent references related studies and indicating the novelty of the study compared to the carried works.
Response 3: Introduction has been revised.
Point 4: Analysis of experimental results
- More details regarding the main experimental results, such as surface characterization and adsorption mechanism sections, must provide in the text.
- All Abbreviations should be defined.
- All Figures should be improved due to the low quality in the manuscript.
- Quantum chemical results should be revised and each considered parameter needs its own unity.
Response 4: this is done. All comments have been considered.
Point 5: General expression issues
The authors ought to go through the manuscript again to eliminate ambiguous expressions which were resulted from complicated sentence structure and grammatical errors.
Respose 5: this is done. all the ambiguous expressions and grammmatical errors have been corrected.
Thank you for your helpful comments. We have revised our paper accordingly and feel that your comments helped clarify and improve our paper
Reviewer 2 Report
The topic of the presented work is highly relevant. Purification of aqueous solutions from contamination with metal salts in the modern world is necessary to maintain an eco-friendly situation in the environment. The development of regenerable adsorbents is a promising area of scientific research.
The authors proposed a new adsorbent for water purification from Cu(II). Among other factors, a wide pH range was studied that expands the possibilities of practical application the adsorbent. The selectivity with respect to Cu (II) ions is an advantage of the presented adsorbent. Theoretical calculations allowed the authors to propose the mechanism of copper ions coordination.
At the same time, there are imperfections in the text and titles of tables/figures, presentation and discussion of the results.
The presented paper can be published after revision.
The comments are listed below:
1. The Section “Materials and methods” should be supplemented with some details of physical-chemical analysis. For example, the heating rate in the TGA, the processing temperature in FTIR and N2 adsorption experiments. Also 13C NMR is listed, but it was not used in the work.
2. Is the observed increase in C, H, and N content consistent with the composition of the functional cover of surface? Is it possible to calculate the number of groups in SiN-imd-py, presented after modification of SiNH2? Does it in correspondence with weight loss obtained by TG analysis?
3. The title of Table 1 “Elemental analysis of SiN-imd-py” should be corrected. This table also presents the results of elemental analysis of SiNH2.
What units are given in Table 1 for the elements content, wt.% or at.% ?
4. It is necessary to show the scale bar on SEM microphotographs.
5. SEM micrographs are strongly charged. It is difficult to discuss the differences in porosity and morphology because of charging effect and different magnification. For example, the large particles are close in morphology, and microphotographs were taken from different regions of the sample. If not, what is the reason of appearance of smaller particles after modification?
6. The title of Figure 1. “FTIR spectra of SiNH2 and Si-Imd-Py (before and after desorption)” does not correspond to the content. The FTIR spectra for SiG, SiNH2 and Si-Imd-Py are presented in Fig.1.
7. The authors confirm the grafting by the appearance of the band at 1440 cm-1 in the FTIR spectrum, but this band has very low intensity. The disappearance of the band at 2932 cm-1 and changes in the spectrum at the 1590-1520 cm-1 region are not discussed.
8. The spelling of the indexes should be checked in the text. For example, in lines 275, 278 and table 3: subscript index in qe.
9. Are there data on thermodynamics for other adsorbents in the literature? It would be nice to compare the obtained values of enthalpy and Gibbs energy with another similar adsorbents, as it was made for adsorption capacities.
10. The pH = 6 is stated in the conclusion to be optimal for adsorption. Why the experiments were performed at pH=7? The statement “The adsorption capacity of Cu(II) ion reached its maximum at a pH between 6 and 7” is presented in the text (line 246)
11. No information presented about the concentrations and pH for experiments with mixed Cu(II), Zn(II), Cd(II) and Pb(II) solution.
12. What is the reason for the decrease in adsorption capacity during recycling experiments? Is a fraction of Cu(II) not washed off from the adsorbent or is it due to the destruction of immobilized ligands? Have experiments been carried out to calculate the material balance?
13. The authors simulate the adsorption process and propose the complexes formed on the surface. Is it possible to estimate what fraction of surface groups that are involved in copper binding, as well as the number of immobilized groups, and the maximum possible capacity of the adsorbent? From a comparison of the obtained data on elemental analysis, IR, and the theoretical mount of Cu(II) that can be adsorbed?
14. It is better to denote the rate constant with a lowercase letter k1 and k2 instead of K1 and K2 to avoid confusion with the equilibrium constant, which is usually denoted by an uppercase letter.
15. The all text, figure captures etc. should be carefully corrected.
Here are only some examples:
a) The title and presentation of Table 5 should be changed, because the thermodynamic parameters are presented only for Cu(II). So, this information should be moved to the title of table.
b) Linу 204: “between 200-800” must be corrected to between 200-800°C
c) In line 525 “C(II)” should be changed to “Cu(II)”.
d) Line 294: it is mentioned in the text that initial concentration of Cu(II) was varied from 10 to 250 mg L−1, while the caption to Fig. 8 shows the interval [Cu (II)] = 10 to 300 mg. L−1.
Author Response
Dear Professor,
Thank you very much for your corrections and recommendations which will certainly improve the quality of the paper.
All the required corrections are deferred below (in red) and in the revised manuscript.
Point 1: The Section “Materials and methods” should be supplemented with some details of physical-chemical analysis. For example, the heating rate in the TGA, the processing temperature in FTIR and N2 adsorption experiments. Also 13C NMR is listed, but it was not used in the work.
Response 1:
- The section "Materials and Methods" is completed with the details of the physicochemical analysis demanded.
- 13C NMR data has been added for the characterization of 2 phenylimidazo[1,2-a]pyridin-3-carbeldehyde.
Point 2: Is the observed increase in C, H, and N content consistent with the composition of the functional cover of surface? Is it possible to calculate the number of groups in SiN-imd-py, presented after modification of SiNH2? Does it in correspondence with weight loss obtained by TG analysis?
Response 2:
- The parameter that allows to characterize the surface grafting is the density of grafted chains per unit of mass or unit of surface of the solid. The measurement of this density is much more complex than it seems. Indeed, the classical techniques of determination of the organic fraction (thermogravimetric analysis or elemental analysis) are not sufficient to obtain a precise measurement but gives an estimate on the grafting and the functionalization of the surface.
- The 13C NMR allows to estimate the proportion of carbon coming from the grafted chains. The grafting rate ng can be calculated directly according to the following formula :
ng= %X*2.47*105/SBET[Mx*nX%Si-28*%X] (molecules/nm2)
Point 3: The title of Table 1 “Elemental analysis of SiN-imd-py” should be corrected. This table also presents the results of elemental analysis of SiNH2.
What units are given in Table 1 for the elements content, wt.% or at.% ?
Response 3:
- The Title of Table 1 was corrected according to the reviewer comment.
- The units are giving in Table 1 for the elements content : wt.%
Point 4: It is necessary to show the scale bar on SEM microphotographs.
Response 4: The scale bar on SEM microphotograohs was added according to the reviewer comment
Point 5: SEM micrographs are strongly charged. It is difficult to discuss the differences in porosity and morphology because of charging effect and different magnification. For example, the large particles are close in morphology, and microphotographs were taken from different regions of the sample. If not, what is the reason of appearance of smaller particles after modification?
Response 5:
- The Figure of SEM images has been modified.
-SEM imaging clearly shows unagglomeration of particles in the case of free silica with particles with an homogeneous surface. Upon functionalization of SiNH2 on silica particles followed by the introduction of a 2-phenylimidazo[1,2-a]pyridin-3-carbeldehyde, a heterogeneous structure with agglomeration of particles is clearly observed.
Point 6: The title of Figure 1. “FTIR spectra of SiNH2 and Si-Imd-Py (before and after desorption)” does not correspond to the content. The FTIR spectra for SiG, SiNH2 and Si-Imd-Py are presented in Fig.1.
Response 6: This was added in the revised version
Point 7: The authors confirm the grafting by the appearance of the band at 1440 cm-1 in the FTIR spectrum, but this band has very low intensity. The disappearance of the band at 2932 cm-1 and changes in the spectrum at the 1590-1520 cm-1 region are not discussed.
Response 7:
* the low intensity of the band on 1440 cm-1 depends on the grafting rate which does not exceed 12%.
* The disappearance of a band around 1580 characteristic of N-H is due to the condensation of the aldehyde CHO with the amine NH2 and thus formation of C=N.
- The characteristic band around 2932 cm-1 is always present at a very low intensity. This band corresponds to the vibration of the C-H bond.
Point 8: The spelling of the indexes should be checked in the text. For example, in lines 275, 278 and table 3: subscript index in qe.
Response 8: This is done
Point 9: Are there data on thermodynamics for other adsorbents in the literature? It would be nice to compare the obtained values of enthalpy and Gibbs energy with another similar adsorbents, as it was made for adsorption capacities.
Response 9: A data on thermodynamics for other adsorbents in the literature was added, The requested references were also added in table 5.
Point 10: The pH = 6 is stated in the conclusion to be optimal for adsorption. Why the experiments were performed at pH=7? The statement “The adsorption capacity of Cu(II) ion reached its maximum at a pH between 6 and 7” is presented in the text (line 246)
Response 10: We thank the reviewer for this observation.The optimal pH values for Cu(II) adsorption were found to be in the pH range of 6-7. This clarification was added in the revised version.
Point 11: No information presented about the concentrations and pH for experiments with mixed Cu(II), Zn(II), Cd(II) and Pb(II) solution.
Response 11:
The details and adsorption conditions have been added in Figure 11.
Point 12: What is the reason for the decrease in adsorption capacity during recycling experiments? Is a fraction of Cu(II) not washed off from the adsorbent or is it due to the destruction of immobilized ligands? Have experiments been carried out to calculate the material balance?
Response 12:
the decrease of adsorption capacity during recycling experiments is due to a small fraction of cu(II) which remains fixed on the adsorbent. never reached 100% regeneration.
Point 13: The authors simulate the adsorption process and propose the complexes formed on the surface. Is it possible to estimate what fraction of surface groups that are involved in copper binding, as well as the number of immobilized groups, and the maximum possible capacity of the adsorbent? From a comparison of the obtained data on elemental analysis, IR, and the theoretical mount of Cu(II) that can be adsorbed?
Response 13:
- From the structure of our material and the theoretical study that we have carried out we can see that the complex formed by coordination bonds between the nitrogen atom N9 and the Cu2+ ions.
- The QTAIM and NCI approach used in this work confirms this mechanism of formation of these coordination bonds between N and Cu2+.
- To estimate the fraction of surface groups involved in copper fixation, as well as the number of immobilized groups, and the maximum possible capacity of the adsorbent it is necessary to use the XPS technique and this is not available to us.
Point 14: It is better to denote the rate constant with a lowercase letter k1 and k2 instead of K1 and K2 to avoid confusion with the equilibrium constant, which is usually denoted by an uppercase letter.
Response 14: This is done
Point 15: The all text, figure captures etc. should be carefully corrected.
Here are only some examples:
a) The title and presentation of Table 5 should be changed, because the thermodynamic parameters are presented only for Cu(II). So, this information should be moved to the title of table.
b) Linу 204: “between 200-800” must be corrected to between 200-800°C
c) In line 525 “C(II)” should be changed to “Cu(II)”.
d) Line 294: it is mentioned in the text that initial concentration of Cu(II) was varied from 10 to 250 mg L−1, while the caption to Fig. 8 shows the interval [Cu (II)] = 10 to 300 mg. L−1.
Response 15: This is done. All comments have been revised and corrected
Thank you for your helpful comments. We have revised our paper accordingly and feel that your comments helped clarify and improve our paper.
Reviewer 3 Report
This study concerns the development of a novel functionalized porous silica for heavy metals removal from water. Nowadays this problem is of major interest because of clean drinking water quantities reduction. Nevertheless, authors should explain which is the novelty of this study. Many other studies using recycled or biobased precursors are exhibiting similar results. Activated bio-carbon is under great interest in this topic. Moreover, the method is not green at all and not environmental friendly. Toluene is of high toxicity and all the other chemicals are not green. If authors could provide sufficient arguments about the novelty and the advantages of this study then this will be a publishable work. Otherwise my opinion is to reject this paper.
Author Response
Point 1:
This study concerns the development of a novel functionalized porous silica for heavy metals removal from water. Nowadays this problem is of major interest because of clean drinking water quantities reduction. Nevertheless, authors should explain which is the novelty of this study. Many other studies using recycled or biobased precursors are exhibiting similar results. Activated bio-carbon is under great interest in this topic. Moreover, the method is not green at all and not environmental friendly. Toluene is of high toxicity and all the other chemicals are not green. If authors could provide sufficient arguments about the novelty and the advantages of this study then this will be a publishable work. Otherwise my opinion is to reject this paper.
Response:
Nowadays, researchers constantly made major progress in functionalization materials technology. This is evident from the large number of paper extensively described in the literature. In this context, all material interacts with its environment via its surface and all interactions with this environment depend on the surface properties of the material. Thus, the same material can have different surface properties depending on how it has been functionalized. Accordingly, our paper describes the functionalization of the silica surface by a pyridine derivative for increases the complexing properties on their surface to improve higher adsorption capacity and selectivity toward Cu(II) compared to others sorbents described in the literature.
The SiN-imd-py adsorbent is important to lead to the high selectivity for Cu2+ which is indeed important. Stating that the paper should be reorganized because SiN-imd-py it is not specific to other heavy metals is not realistic given that the establishment of such ideal adsorbent is still a research objective for many groups and has not yet been reported. It is therefore useful to remind the general input of this work summarized as follows:
- The adsorption capacity of our adsorbent (SiN-imd-py) towards Cu2+ is much more important (≈100 mg/g) compared to several recent adsorbents previously known and published in specialized journals.
Round 2
Reviewer 3 Report
Authors provided a supported argument about the functionalized silicas and the way they work for heavy metals ramoval. We also agree that 100mg/g capacity is of high scores. They also changed the abstract section and the term environmetally friendly process is dissappeared. Under the spirit that every scientific research on heavy metals removal from aqueous solutions could be important I agrree for this revised version of the paper to be published.